# Fusion Algorithm of the Improved A* Algorithm and Segmented Bézier Curves for the Path Planning of Mobile Robots

Rongshen Lai [1],*, Zhiyong Wu [1], Xiangui Liu [1],* and Nianyin Zeng [2]

1 School of Mechanical and Automotive Engineering, Xiamen University of Technology, Xiamen 361024, China
2 Department of Instrumental and Electrical Engineering, Xiamen University, Xiamen 361005, China
* Correspondence: 2016000053@xmut.edu.cn (R.L.); 2010110819@xmut.edu.cn (X.L.)

**Abstract:** In terms of mobile robot path planning, the traditional A* algorithm has the following problems: a long searching time, an excessive number of redundant nodes, and too many path-turning points. As a result, the shortest path obtained from planning may not be the optimal movement route of actual robots, and it will accelerate the hardware loss of robots. To address the aforementioned problems, a fusion algorithm for path planning, combining the improved A* algorithm with segmented second-order Bézier curves, is proposed in this paper. On the one hand, the improved A* algorithm is presented to reduce unnecessary expansion nodes and shorten the search time, which was achieved from three aspects: (1) the traditional 8-neighborhood search strategy was adjusted to 5-neighborhood according to the orientation of the target point relative to the current node; (2) the dynamic weighting factor of the heuristic function was introduced into the evaluation function of the traditional A* algorithm; and (3) the key node extraction strategy was designed to reduce the redundant nodes of the optimal path. On the other hand, the optimal path planned by the improved A* algorithm was smoothed using segmented second-order Bézier curves. The simulation results show that the improved A* algorithm can effectively reduce the search time and redundant nodes and the fusion algorithm can reduce the path curvature and path length to a certain extent, improving path safety.

**Keywords:** mobile robot; path planning; improved A* algorithm; Bézier curves; fusion algorithm

## 1. Introduction

In recent years, mobile robots, such as handling robots, sorting robots, and medical robots, have been widely used in industrial manufacturing, logistics, and medical fields. With the increasing popularity of mobile robots, path planning, that is, finding an optimal path from the starting point to the target point without collision for a specific obstacle environment, has become a key aspect affecting their functions and performance [1,2]. According to the mastery of the obstacle environment, path planning algorithms are usually divided into two categories, global algorithms and local algorithms. The former mainly includes Dijkstra [3], RRT (Rapidly exploring Random Tree) [4,5], and A* [6,7], and the latter mainly includes Dynamic Window Approach (DWA) [8,9], Artificial Potential Field Approach [10,11], Particle Swarm Algorithm [12,13], and Ant Colony Algorithm [14]. Among these algorithms, Dijkstra's algorithm can find the global optimal path, but it is difficult to complete planning in a short period of time, due to the large number of nodes; RRT algorithm has a fast search speed, but the search accuracy is low, the path is not smooth, and it is difficult to obtain the optimal path; and the A* algorithm, widely used in global path planning, introduces a heuristic function to the evaluation function of Dijkstra's algorithm, which accelerates the search speed and improves the search efficiency.

However, the traditional A* algorithm has some deficiencies, such as a long search time, too many redundant nodes, and too many path-turning points. For this reason, scholars have proposed a series of optimization methods. The authors of [15,16] selectively

expanded the neighborhood to reduce unnecessary expansion nodes, which fits the obstacles and overcomes the defects of the path obtained by the traditional A* algorithm. The authors of [17,18] improved the heuristic function of the A* algorithm in order to shorten the search time and reduce the number of nodes traversed. The authors of [19] added location information to the evaluation function of the traditional A* algorithm to shorten the path length and reduce the running time. The authors of [20] weighted the evaluation function of the traditional A* algorithm to reduce the search step length and the search time of path planning. The authors of [21–23] proposed an improved A* algorithm combined with the JPS jump point method in order to reduce the number of extended nodes and improve the search efficiency. The authors of [24,25] used abi-directional A* algorithm to search from both positive and negative directions simultaneously to improve the search efficiency. When the mobile robot faces a complex environment, using only a single global path planning algorithm, such as the traditional A* algorithm, cannot effectively and efficiently complete the path planning task and must combine the global algorithm with the local algorithm. The authors of [26,27] combined the A* algorithm with the artificial potential field method to achieve the path planning problem in dynamic environments, but the local path planning results of the artificial potential field method were less satisfactory. The authors of [28,29] proposed a hybrid algorithm based on improved A* algorithm and dynamic window method to overcome the shortcomings of traditional path inflection points, but did not consider the processing of temporary obstacles. The authors of [30,31] proposed the combination of reinforcement learning with the A* algorithm to reduce the computational complexity in the path planning process. The authors of [32,33] combined genetic algorithm with the A* algorithm to reduce the path length. The authors of [34] combined the ant colony algorithm with the A* algorithm to reduce the expansion nodes and improve the search efficiency when searching the path. The authors of [35,36] combined the A* algorithm with the TEB algorithm to make the planned path smoother and improve the safety of robot movement. The authors of [37] combined the improved D* Lite algorithm with sub-objective based hybrid path planning, which both shortens the path of global path planning and has a safe and comfortable obstacle distance.

In summary, for the path planning problem of mobile robots in complex environments, the traditional A* algorithm has the shortcomings of a long search time and too many redundant nodes. Based on the current research on the improvement of the traditional A* algorithm, this paper proposes a fusion algorithm combining the improved A* algorithm with segmented second-order Bézier curves. Firstly, the traditional 8-neighborhood search strategy is adjusted to 5-neighborhood according to the orientation of the target node relative to the current node, which reduces the number of extended nodes and improves the search efficiency. Secondly, the dynamic weighting factor of the heuristic function is introduced into the traditional evaluation function, which makes the role of the heuristic function change dynamically, that is, when the current node is far from the target node, the algorithm focuses on the search speed, and when the current node is close to the target node, the algorithm focuses on the search accuracy; after that, the key nodes are extracted from the planned path to retain key nodes and reduce unnecessary redundant nodes. Finally, the obtained optimal path is smoothed using a segmented second-order Bézier curve to obtain a smoother and collision-free path. The effectiveness of the proposed fusion algorithm is verified through simulation experiments on a 2D raster map with multiple obstacles.

The main contributions and uniqueness of this research include the following four aspects. (1) A 5-neighborhood search strategy is proposed to optimize the search path, shorten the algorithm execution time and reduce the redundant nodes. (2) The dynamic weighting factor is added to the heuristic function of the traditional A* algorithm, so that the algorithm can automatically adjust the search focus during the search process, solving the common problem of long algorithm running time of the traditional A* algorithm. (3) A key node extraction strategy is proposed to solve the problem of many redundant nodes in the original global path. (4) A fusion algorithm combining the improved A* algorithm

and segmented Bézier curves is proposed to smooth the local paths and make the routes smoother and safer.

The remainder of this paper is organized as follows. Section 2 describes the improved A* algorithm based on the traditional A* algorithm. Section 3 introduces the basic performance of the Bézier curves. Section 4 proposes the fusion algorithm based on the improved A* algorithm and the segmented second-order Bézier curve. Section 5 analyzes the performance of the fusion algorithm. Section 6 discusses the advantages and disadvantages of the proposed fusion algorithm. Section 7 concludes the paper and proposes future research directions.

## 2. Improved A* Algorithm

### 2.1. Environment Building

In the study of mobile robot path planning, there are mainly three methods for constructing map environments: topological maps [38], geometric maps [39], and raster maps [40]. Of these, raster maps are simple and intuitive, easy to build, represent and save, and can effectively compress data, which is also convenient for calculating the area, length, turning direction, and concavity. Therefore, in this paper, raster maps were used to build the working environment of mobile robots, in which the black raster indicates the obstacle area that the robot cannot reach and the white raster indicates the free moving area that the robot can reach, as shown in Figure 1.

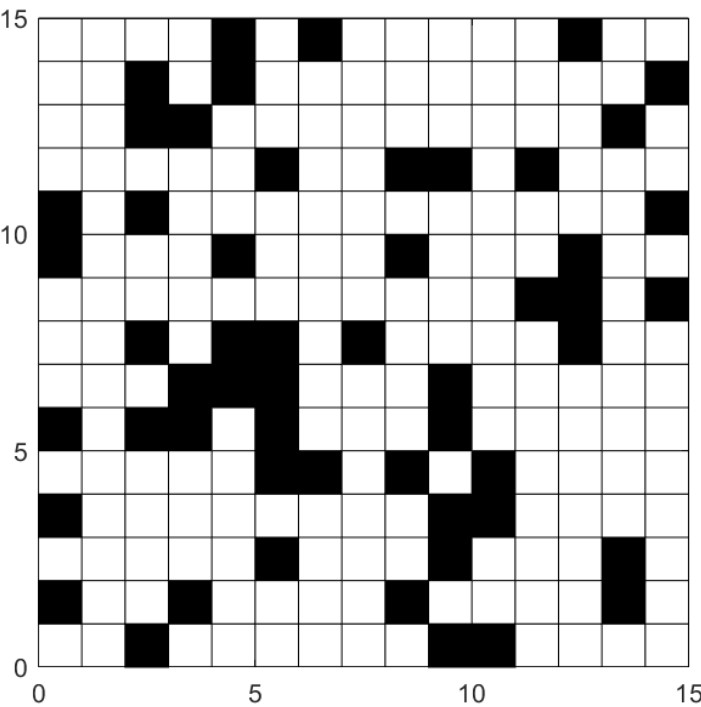

**Figure 1.** Raster environment.

### 2.2. Improved A* Algorithm

#### 2.2.1. Traditional A* Algorithm

The A* algorithm is a common global path planning algorithm for mobile robots. It adds the estimated movement cost from the current node to the target node based on Dijkstra's algorithm, and the basic process of A* algorithm is as follows: starting from the original node and expanding to the surrounding area; selecting a node with the minimum total movement cost as the starting node for the next search; and setting the previous node as the father node of the current node. The A* algorithm combines the advantages

of Dijkstra's algorithm and BFS (breadth-first traversal) algorithm [41], and its evaluation function is shown as in Equation (1).

$$F(n) = G(n) + H(n) \tag{1}$$

where $F(n)$ denotes the total movement cost from the current node to the target node; $G(n)$ denotes the actual movement cost from the current node to the starting node; and $H(n)$ denotes the estimated movement cost from the current node to the target node. The commonly used methods for calculating $H(n)$ are Manhattan distance, Euclidean distance, and Chebyshev distance. In order to be more consistent with the actual distance when the robot moves, the Euclidean distance was used in this paper, and the calculation formula is shown in Equation (2).

$$H(n) = \sqrt{(x_2 - x_1)^2 + (y_2 - y_1)^2} \tag{2}$$

where $(x_1, y_1)$, $(x_2, y_2)$ denote the coordinates of the current node and the node whose distance is to be sought, respectively. The specific steps of the A* algorithm are as follows.

(1) Initialize the openList list as well as the closeList list, and add the starting node to the openList list.
(2) Iterate through the openList list and find the node $n$ with the smallest total movement cost $F$ value according to the evaluation function as the current node to be processed, and add node $n$ to the closeList list.
(3) Check the eight neighboring nodes of node $n$. In turn, if the node is an unreachable obstacle area or already exists in the closeList list, skip the node; otherwise perform the following operations.

　　If the node is not in the openList list, add it to the openList list and set node $n$ as the father node of the node.

　　If the node already exists in the openList list, check whether the total movement cost of node $n$ to reach the node is closer, and if so, set its father node to node $n$ and calculate the total movement cost to update the node.

　　The search is stopped when the endpoint is added to the openList list, at which point the endpoint is reachable and the path is found, or when the end point is not yet reachable, the path is not found, but the openList list is empty and the path finding has failed.

(4) Repeat steps (2) and (3).
(5) Starting from the target node, find the father node of each node in turn, and connect to generate a complete optimal path.

　　Although the traditional A* algorithm can find a shortest path from the starting node to the target node faster, but the planned path still has many redundant nodes and fold points, resulting in a low search efficiency, and even the phenomenon of crossing obstacles. For this reason, this paper proposes to improve the A* algorithm based on the traditional A* algorithm by optimizing three aspects, i.e., search strategy, evaluation function, and key node extraction.

### 2.2.2. Search Strategy

　　The search strategy of the traditional A* algorithm is to start from the central node, spread to the surrounding 8 neighborhoods in turn to search for the next node to be expanded, and then select the expanded node based on the evaluation function calculation. According to the orientation of the target node relative to the current node, the 3 neighborhoods that deviate from the orientation of the target node are discarded and the search strategy is adjusted to 5 neighborhoods, as shown in Figure 2. In the figure, node ⑤ is taken as the central node, and the discarded 3areexpanded and determined according to the orientation of the target node: when the target node is located at the bottom right of the central node (Figure 2a), nodes ①, ②, and ④ are discarded, and only nodes ③, ⑥, ⑦, ⑧, and ⑨, which are not obstacles, are put into the set of nodes to be expanded, and

considering the problem of crossing obstacles that occurs in the traditional A* algorithm, when nodes ⑥ and ⑧ are obstacles, node ⑨ is set as inaccessible. Similarly, when the target point is located in the lower left, upper right and upper left, the 3 neighborhoods are rounded off according to the orientation of the target node, and the nodes which are not obstacles in the remaining nodes are placed in the set of nodes to be expanded, and the problem of crossing obstacles is considered, after which the expanded nodes are selected by comparing the evaluation function values to obtainan optimal path that reaches the target node faster.

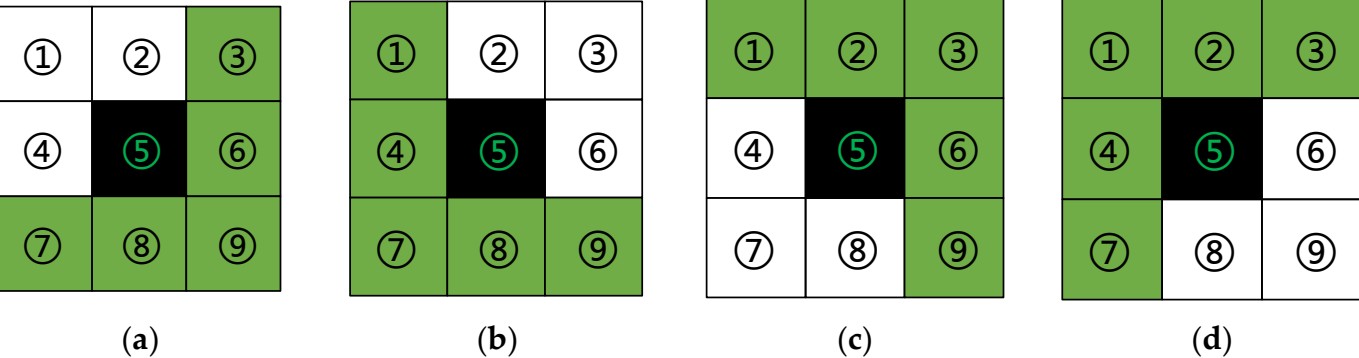

**Figure 2.** Improved 5-neighborhood search strategy. (**a**) Bottom right target, (**b**) bottom left target, (**c**) top right target and (**d**) top left target.

The experimental verification shows that, although the improved A* algorithm using the 5-neighborhood search strategy achieves the purpose of reducing redundant extension nodes, the effect on improving the search speed is not very significant, so an improvement scheme for the evaluation function of the traditional A* algorithm is proposed in the next section.

### 2.2.3. Evaluation Function

The evaluation function of the traditional A* algorithm is a combination of the actual movement cost $G(n)$ from the starting node to the current node and the estimated movement cost $H(n)$ from the current node to the target node. In this paper, a dynamic heuristic factor (Equation (3)) was added to the heuristic function of the traditional A* algorithm to strengthen the role of $H(n)$ when the current node is far away from the target node to speed up the search and weaken the role of $H(n)$ when the current node is close to the target node to improve the search accuracy.

$$F(n) = G(n) + W * H(n) \tag{3}$$

where $W$ is a weighting factor determined from the sum of the length and width of the map and the Manhattan distance between the current node and the target node, constructed as follows.

$$d(x, y) = |x_1 - x_2| + |y_1 - y_2| \tag{4}$$

$$W = 1 + d(x, y)/(m + n) \tag{5}$$

In Equation (4), $(x_1, y_1)$, $(x_2, y_2)$ denote the coordinates of the current node and the nodewhose distance is to be sought, respectively. In Equation (5), $m$ and $n$ denote the length and width of the raster map, respectively.

On the basis of using the 5-neighborhood search strategy and adding dynamic weighting factors to the heuristic function of the A* algorithm, the number of expansion nodes decreases and the search time reduces significantly, as shown in Figure 3. However, the problem of too many redundant nodes in the optimal path after planning is not improved, and a further extraction of key nodes is needed.

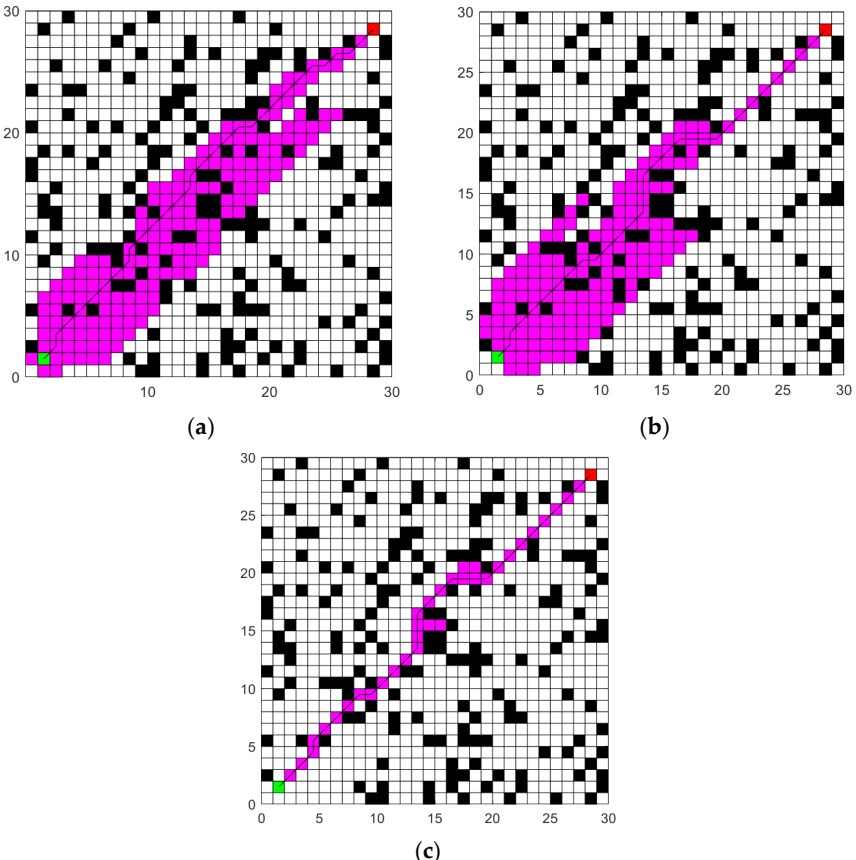

**Figure 3.** Comparison of expansion nodes before and after optimization. (**a**) Traditional A* algorithm, (**b**) 5-Neighborhood search strategy, and (**c**) 5-Neighborhood search strategy and with dynamic weighting factor.

### 2.2.4. Key Node Extraction

For the problem that there are still too many redundant nodes and redundant sections in the path after planning with A* algorithm, the key node extraction algorithm is proposed to eliminate the redundant nodes in the path and retain the key nodes so that the total number of nodes after optimization is smaller. The specific steps of the key node extraction algorithm are as follows, as shown in Figure 4.

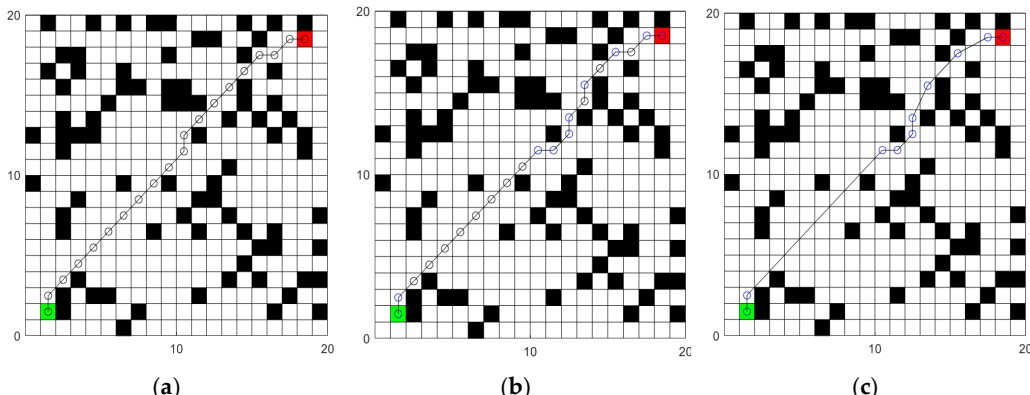

**Figure 4.** Key node extraction steps. (**a**) Original planned path, (**b**) extraction of key nodes, and (**c**) connecting key nodes.

(1)   Create a keypoint list of key nodes, and join the starting node and the target node into the keypoint list.

(2)   In the planned path, begin with the start node *m*, connect node *m* and node *m*+2 of the adjacent nodes (*m*, *m*+1, *m*+2), and judge whether the connected path passes through the obstacle area or not.

(3)   If the connected path passes through the obstacle area, node *m*+1 is the keypoint of the path. The key node *m*+1 is stored in the keypoint list, and node *m*+1 is used as the new starting node *m*, and return to step (2); if the connected path does not pass through the obstacle area, node *m*+1 is the redundant node on the path, and node *m*+1 is removed from the planned path, and node (*m*+2,*m*+3) is used as the new node (*m*+1,*m*+2), and return to step (2).

(4)   When the target node is connected, the key node extraction ends and the key nodes in the keypoint list are connected to obtain the final path.

## 3. Bézier Curve

### 3.1. General Introduction

Bézier curves require only a few control points to generate more complex smoothing curves. In this paper, the optimal planned path with the improved A* algorithm was smoothed by using Bézier curves. Usually, *n*+1 control points are defined to form an *n*th-order Bézier curves, and the expression is shown in Equation (6).

$$P(t) = \sum_{i=0}^{n} P_i B_{i,n}(t), t \in [0,1] \tag{6}$$

In Equation (6), $P_i$ and $t$ are the coordinate values and parameters of the control points, respectively, and $B_{i,n}(t)$ is the Bernstein polynomial, whose expression is shown in Equation (7).

$$B_{i,n}(t) = C_n^i t^i (1-t)^{n-i}, i = 0,1,\ldots,n \tag{7}$$

In Equation (7), $C_n^i$ is the quadratic term coefficient and n is the order of the Bézier curves. The commonly used parametric expressions for Bézier curves are shown as follows.

(1)   Second-order Bézier curve.

$$P(t) = P_0(1-t)^2 + 2P_1(1-t)t + P_2 t^2, t \in [0,1] \tag{8}$$

(2)   Third-order Bézier curve.

$$P(t) = P_0(1-t)^3 + 3P_1(1-t)^2 t + 3P_2(1-t)t^2 + P_3 t^3, t \in [0,1] \tag{9}$$

According to Equations (8) and (9), the second-order Bézier curve passes through the first control point $P_0$ (*t* =0) and the third control point $P_2$ (*t* =1); the third-order Bézier curve passes through the first control point $P_0$ (*t* =0) and the fourth control point $P_3$ (*t* =1).

The tangent vector of the curve at the endpoint is shown as follows.

(1)   Second-order Bézier curve.

$$P\prime(t) = -2P_0(1-t) + 2P_1(1-2t) + 2P_2 t, t \in [0,1] \tag{10}$$

(2)   Third-order Bézier curve.

$$P\prime(t) = -3P_0(1-t)^2 + 3P_1\left(3t^2 - 4t + 1\right) + 6P_2(1-t)t + 3P_3 t^2, t \in [0,1] \tag{11}$$

The curvature of the Bézier curve at any point is shown in Equation (12).

$$k(t) = \frac{x'(t)y''(t) - y'(t)x''(t)}{\left(x'^2(t) + y'^2(t)\right)^{\frac{3}{2}}} \tag{12}$$

The curvature of the Bézier curve at the starting point is shown in Equation (13).

$$k(0) = \frac{3}{4} * \frac{|(P_1 - P_0) * (P_2 - P_1)|}{|P_1 - P_0|^2} \tag{13}$$

The Bézier curve has the invariant property of affine transformation, that is, arbitrary rotation or translation of the curve position will not change the shape of the curve. According to the curvature formula of Equation (12), the curve is continuous when $x'(t)$ and $y'(t)$ are not 0 at the same time; but when $x'(t)$ and $y'(t)$ are 0 at the same time, the curve cannot reach the subsequent control point, which contradicts the definition of the Bézier curve, so the curvature of the trajectory determined by either second-order Bézier curve or third-order Bézier curve is continuous everywhere.

### 3.2. Segmented BézierCurve

Due to the many twists and turns and large corners of the planned path in the raster map, the mobile robot keeps making sharp turns and even stopping its movement in the middle of driving. Such unstable motion not only causes the mobile robot to operate less efficiently, but also leads to an increase in power consumption and wear and tear. Therefore, many researchers have started to use traditional Bézier curves to smooth the optimal paths obtained from planning.

Since the numerical stability of higher-order Bézier curves is poor, in this paper, segmented second-order Bézier curves were used for the smoothing of paths by judging whether the slope of the line connecting adjacent nodes satisfies a certain slope threshold interval. The detailed processes are as follows.

(1) Connect the neighboring nodes (*m*, *m*+1) and (*m*+1, *m*+2) in the path and judge whether the slope of the connected line meets the slope threshold interval; if it does, add the node (*m*, *m*+1, *m*+2) to the array to be smoothed and perform the second-order Bézier curve smoothing; if it does not meet the slope threshold interval, set the node *m*+1 as the current starting point.
(2) Repeat step (1) to traverse the path nodes in turn until all the path nodes are traversed.

## 4. Fusion Algorithm

In this paper, the fusion algorithm combining the improved A* algorithm with segmented Bézier curves is proposed, where the path planning using the improved A* algorithm is implemented first and then the segmented second-order Bézier curves is used to smooth the planned path, so that the planned path has fewer redundant nodes and the path is smoother. The fusion algorithm is described in detail as follows, part of the pseudocode is shown in Algorithm 1, and the specific processes are shown in Figure 5.

(1) Start the program.
(2) Initialize the openList and closeList, and generate the raster map.
(3) Determine whether the openList is empty or not, and if it is, execute step (12).
(4) Find the node with the lowest total movement cost from the openList based on the weighted evaluation function, remove it from the openList, and add it to the closeList.
(5) Determine whether the current node is the target node; if it is, execute (11).
(6) Determine the expansion nodes using the 5-neighborhood search strategy according to the target node orientation and obstacle distribution.
(7) Determine whether the expansion node is an obstacle; if it is, replace it with another expansion node.
(8) Judge whether the expansion node is in the openList; if it is not, add the node to the openList and execute (5).
(9) Determine whether the actual movement cost *G* of the expansion node is smaller; if it is, update the expansion node; if it is larger, execute (4).
(10) Update the value of *G* of the expansion node and its parent node, and then execute (6).

(11) Connect the nodes in the closeList to obtain the optimal path, then use the key point extraction strategy for extraction, and finally use the segmented Bézier curve for the smoothing operation.

(12) End the program.

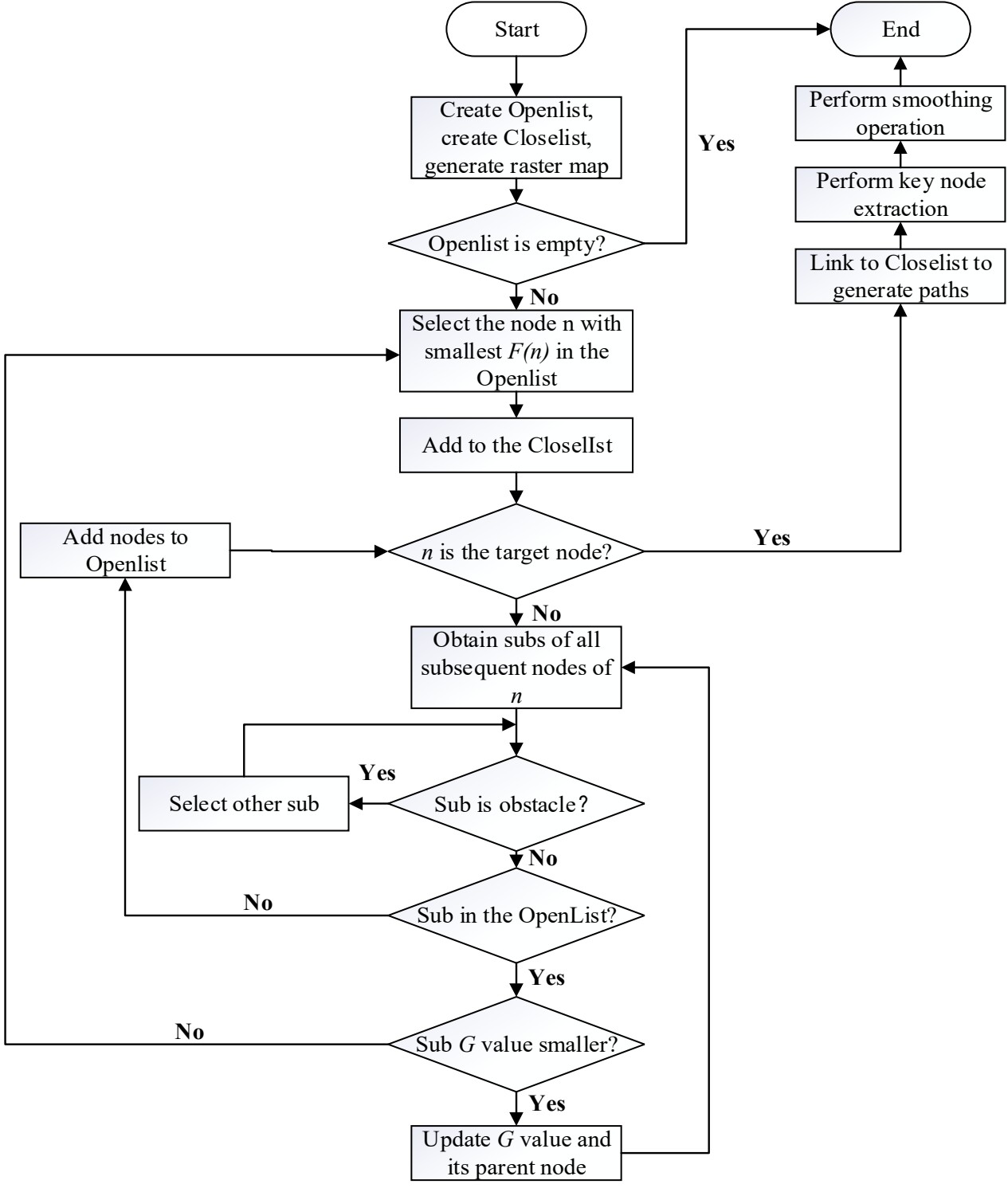

**Figure 5.** The flowchart of the fusion algorithm.

---

**Algorithm 1** Fusion algorithm combining improved A* with segmented Bézier

---

1: **Main()**
2:      **Initialize()**
3:      **Astar()**
4:      **Bezier()**
5: **End**
6: **function** Initialize(M,N)
7:      **Create Map(M,N)**
8:      $m \leftarrow 0$
9:      $O \leftarrow \varnothing$
10:      $C \leftarrow \varnothing$
11: $O.Update(m)$
12: **end function**
13:
14: **function** Astar(*start,target,flag=true*)
15:      **while** *flag* **do**
16:          **for** $m \epsilon O$ **do**
17:              $n \leftarrow m.min()$
18:          **end for**
19:          $C.Update(n)$
20:          $O.Delete(n)$
21:          **for** $m \epsilon n.subs()$ **do**
22:              **if** $m=obstacle$ **or** $m \epsilon O$ **then**
23:                  **Skip**
24:              **else**
25:                  $O.Update(m)$
26:                  $n \rightarrow m.father$
27:              **end if**
28:          **end for**
29:          **if** $target \epsilon O$ **or** $O \rightarrow \varnothing$ **then**
30:              flag=false
31:              **Break**
32:          **end if**
33:      **end while**
34:      **for** $m \epsilon path$ **do**
35:          **KeyPoint(m,m+1,m+2)**
36:      **end for**
37:      path
38: **end function**
39:
40: **function** Bezier(*path*)
41:      **for** $m \epsilon path$ **do**
42:          **SlopeComp(m,m+1,m+2)**
43:      **end for**
44:      return path
**45: end function**

---

## 5. Simulation and Analysis

In order to verify the performance of the improved A* algorithm and the fusion algorithm proposed in this paper, simulation comparison experiments are conducted in this section under the operating system Windows 10 and processor i5-10400 environment, and the results of the simulation comparison experiments are analyzed.

### 5.1. Simulation Experiments and Analysis of Improved A* Algorithm

To verify the effectiveness of the improved A* algorithm, different environments were constructed for simulation experiments. The simulation experiment environment is a two-dimensional raster map with black grids as obstacle nodes, white grids as pass-

able path nodes, green grids as the starting node, and red grid as the target node. The experiments were conducted in three different environments, which are $20 \times 20$, $40 \times 40$, and $60 \times 60$ size, with an effective obstacle rate of 20% in the two-dimensional raster map environment. The results of the simulation experiments are shown in Figure 6, and further comparisons in terms of path length, operation time, total nodes, path length reduction ratio, operation time reduction ratio, and total nodes reduction ratio are shown in Table 1 and Figure 7.

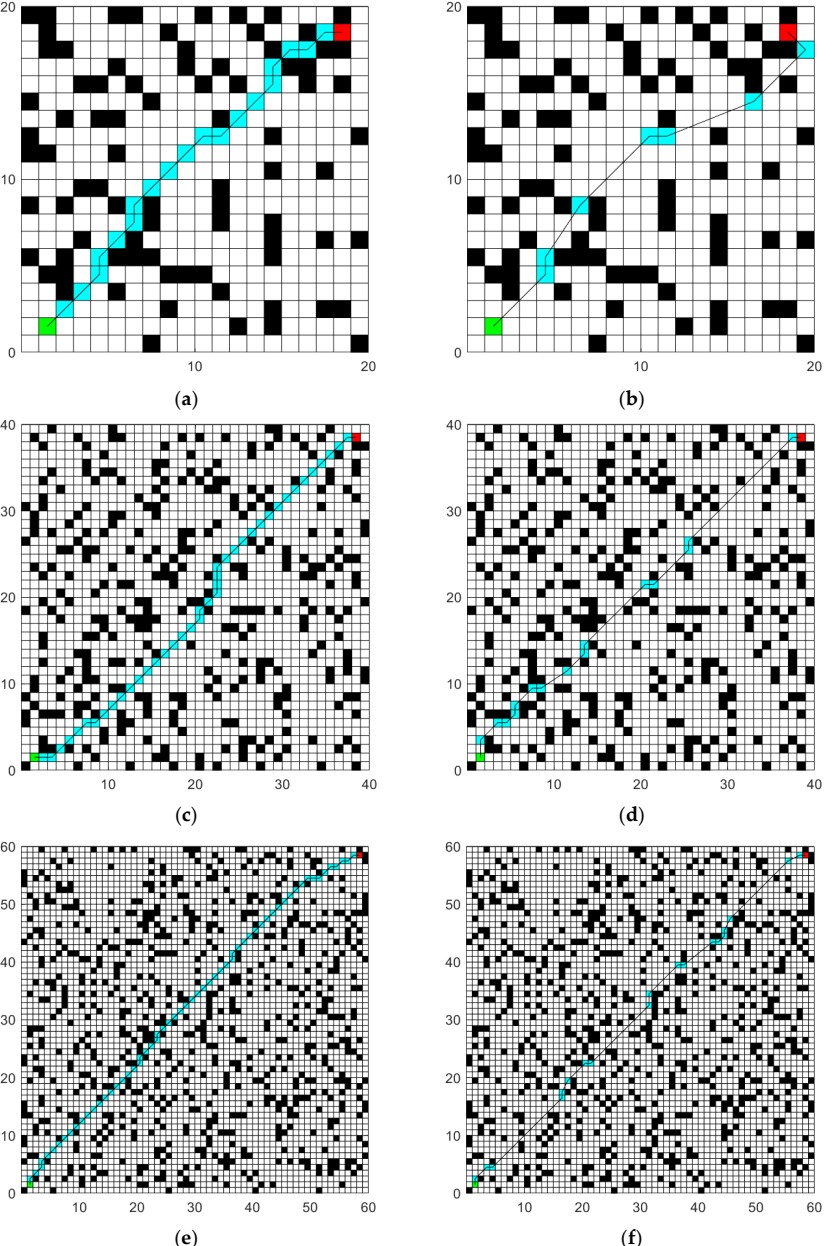

**Figure 6.** Comparison of the experiment results between the improved A* and traditional A* algorithms.(**a**) Traditional A* algorithm ($20 \times 20$), (**b**) improved A* algorithm ($20 \times 20$), (**c**) traditional A* algorithm ($40 \times 40$), (**d**) improved A* algorithm ($40 \times 40$), (**e**) traditional A* algorithm ($60 \times 60$), and (**f**) improved A* algorithm ($60 \times 60$).

**Table 1.** Performance comparison of the improved A* and traditional A* algorithms.

| Map Size | Algorithm | Path Length | Operation Time | Total Nodes | Pathlength Reduction Ratio | Operation Time Reduction Ratio | Total Nodes Reduction Ratio |
|---|---|---|---|---|---|---|---|
| 20 × 20 | Traditional A* | 25.7989 | 0.1151 | 21 | −2.9% | 67.4% | 57.1% |
| | Improved A* | 26.5471 | 0.0375 | 9 | | | |
| 40 × 40 | Traditional A* | 54.6691 | 0.3437 | 42 | −0.6% | 78.5% | 59.5% |
| | Improved A* | 55.0319 | 0.0739 | 17 | | | |
| 60 × 60 | Traditional A* | 83.5391 | 0.8512 | 63 | −0.4% | 89.1% | 66.7% |
| | Improved A* | 83.9225 | 0.0926 | 21 | | | |

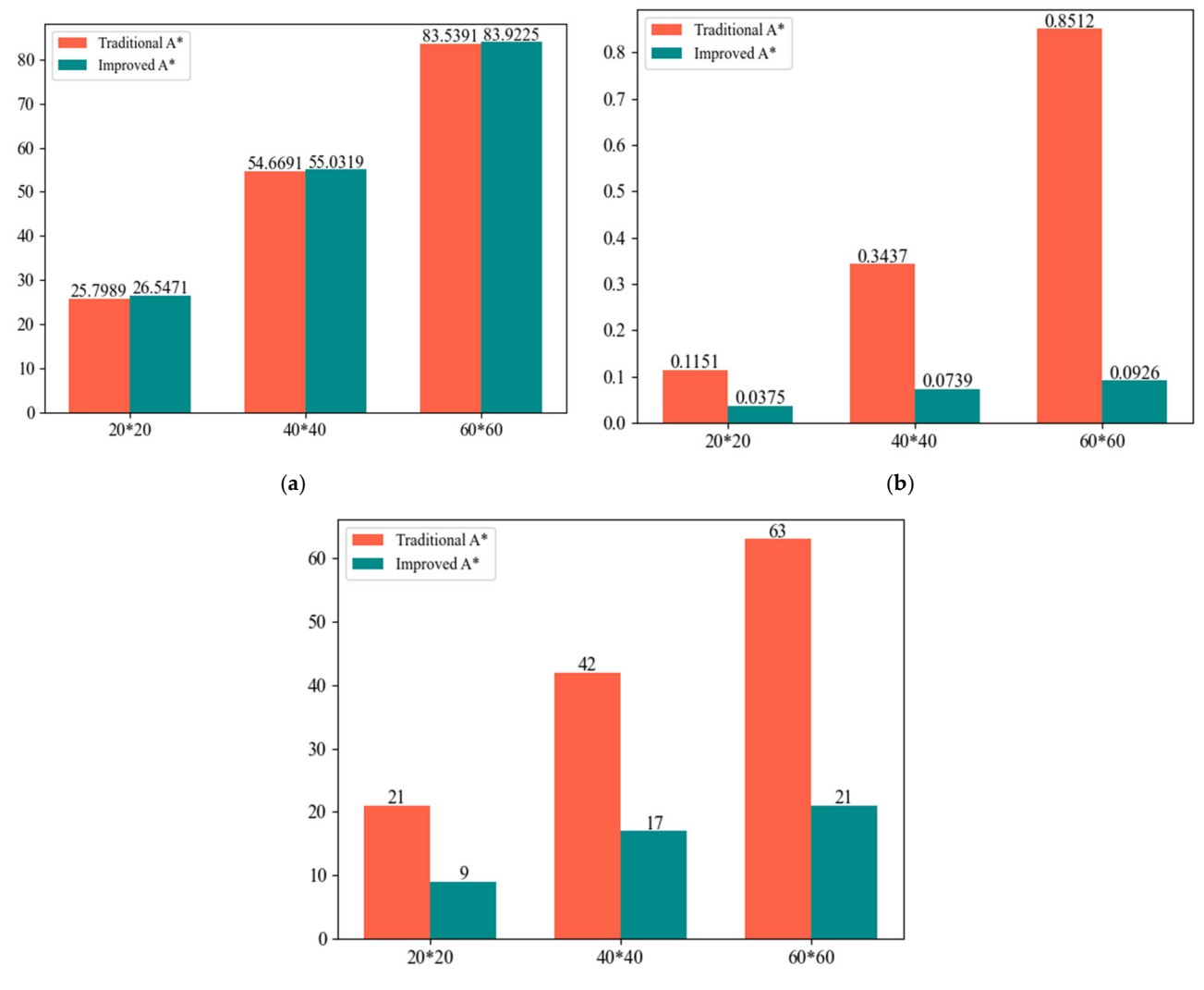

**Figure 7.** Performance comparison of the improved A* and traditional A* algorithms. (**a**) Optimal path length, (**b**) Operation time, (**c**) Total nodes of the optimal path.

According to Figure 6a–f and Table 1, it can be seen that, under the same obstacle environment, the optimal path length obtained by the improved A* algorithm planning slightly increases compared with that of the traditional A* algorithm, but the algorithm running time and the optimal path nodes decrease significantly due to the introduction of dynamic weighting factors to the traditional A* algorithm evaluation function and use of the key point extraction strategy. According to Figure 7b,c and Table 1, it can be seen that, in the environment with the grid size of 20 × 20 and the effective obstacle rate of 20%, the algorithm running time and the total number of optimal path nodes decrease by 67.4% and 57.1%, respectively; in the environment with the grid size of 40 × 40 and the effective

obstacle rate of 20%, the algorithm running time and the total number of optimal path nodes decrease by 78.5% and 59.5%, respectively; and in the environment with a grid size of 60 × 60 and an effective obstacle rate of 20%, the running time and the total number of optimal path nodes decreased by 89.1% and 66.7%, respectively. In summary, although the improved A* algorithm slightly increases the path length, the decrease in the number of nodes and the operation time consumed by the improved A* algorithm are more obvious compared with the traditional A* algorithm as the size of raster map continues to grow.

### 5.2. Simulation Experiments and Analysis of the Fusion Algorithm

The fusion algorithm combines the improved A* algorithm with the segmented second-order Bézier curves to make the curves smoother and further reduces the path length. The simulation experiment environment of the fusion algorithm in this paper is a two-dimensional raster map of size 30 × 30 and 50 × 50 with an effective obstacle rate of 25%. The results of the simulation experiments are shown in Figure 8.

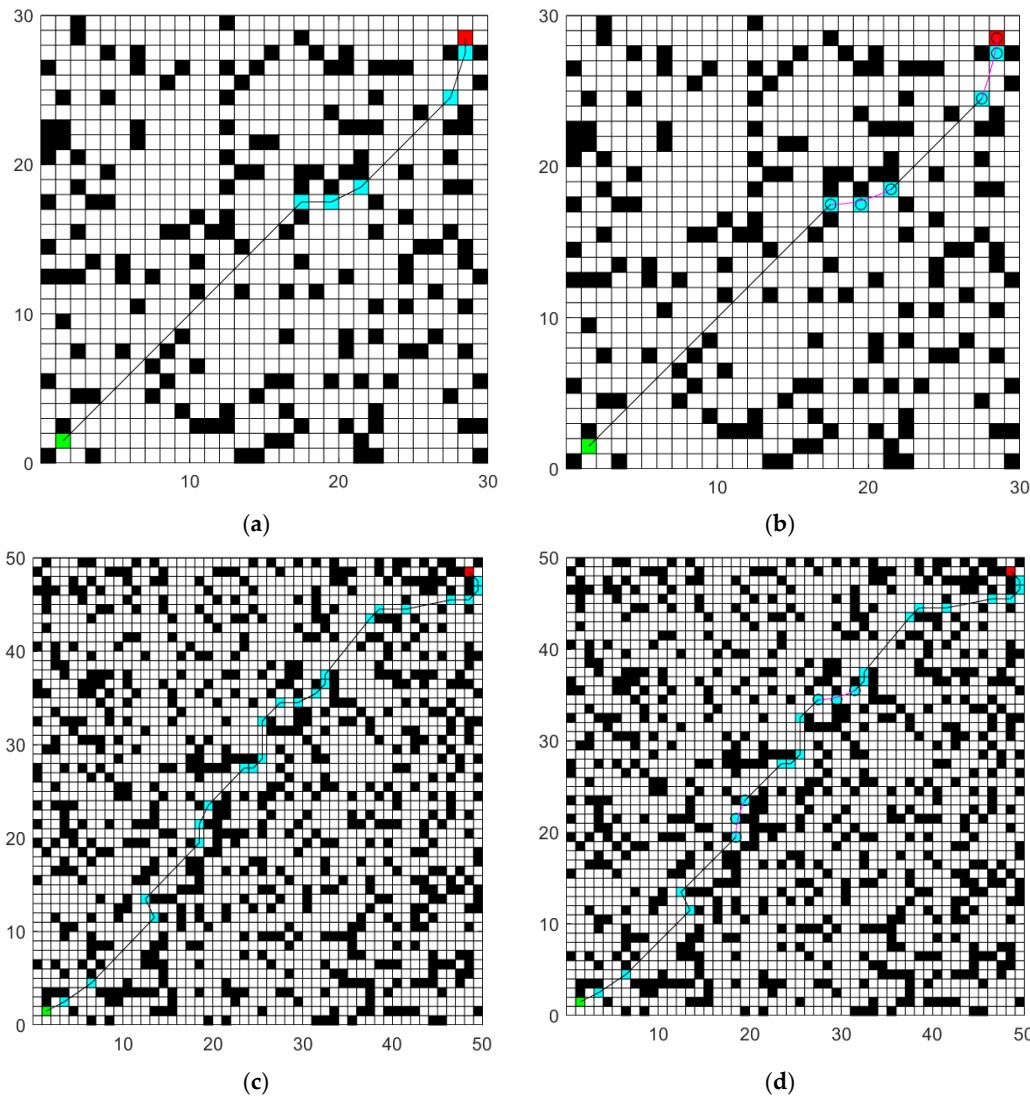

**Figure 8.** Simulation experiment results of the fusion algorithm.(**a**) Before smoothing (30 × 30), (**b**) after smoothing (30 × 30), (**c**) before smoothing (60 × 60), and (**d**) after smoothing (60 × 60).

According to Figure 8 and Table 2, it can be seen that the paths planned by the improved A* algorithm still have some points with large inflection, and the path smoothing

operation using segmented second-order Bézier curves can reduce the path curvature and path length to a certain extent.

**Table 2.** Performance comparison of the fusion algorithms.

| Map Size | Algorithm | Path Length |
|---|---|---|
| $30 \times 30$ | Improved A* algorithm | 39.5110 |
| | Fusion algorithm | 39.4833 |
| $50 \times 50$ | Improved A* algorithm | 75.4002 |
| | Fusion algorithm | 75.3250 |

## 6. Discussion

The search neighborhood, evaluation function, and the total number of nodes after planning of the traditional A* algorithm were optimized in this paper, which has significant effects on shortening the running time of the algorithm and reducing the total number of nodes on the optimal path, and the subsequent application of segmented Bézier curves also plays an important role in improving the smoothness of the optimal path. In the actual working environment, the shortening of the algorithm running time can effectively improve the operation completion efficiency; the reduction in the total number of nodes on the optimal path and the improvement of the path smoothness can reduce the frequent acceleration and deceleration of the robot, improve the operation stability of the mobile robot, and extend the life of the mobile robot to a certain extent.

However, it was found that the proposed algorithm still has shortcomings that need further optimization, such as when the A* algorithm changes the search neighborhood from the traditional 8-neighborhood to 5-neighborhood, it may encounter the dead-end position as shown in Figure 9.In that case, the algorithm in this paper cannot escape the "trap", but as was verified by the simulation experiments, the probability of trapping in a random environment is low (about 10%), yet a relatively common problem in a realistic environment. Therefore, the5-neighborhood search can be used as the first option and the 8-neighborhood search as the alternative option, and then decide whether to enable the alternative option by judging whether the 5-neighborhood search enters the "trap" or not. A further investigation of this problem will be carried out in subsequent work.

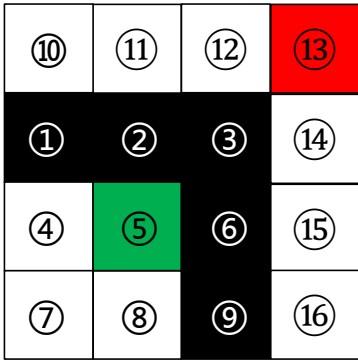

**Figure 9.** Dead-end position.

From a comparison perspective, the proposed fusion algorithm was compared with the traditional A* algorithm and the improved A* algorithm, and the improved A* algorithm was compared with the traditional A* algorithm. The comparison results show that the fusion algorithm is able to optimize the path planning problem, which illustrates the significance of this research. However, the proposed method should also be further compared with other recently proposed algorithms to solve the path planning issue of mobile robots.

Moreover, the present research mainly considered randomly generated raster maps, but more complex obstacles, such as U-shaped obstacles, were not analyzed to verify the effectiveness of the proposed method.

## 7. Conclusions

Path planning is the key link for a mobile robot to complete a given task. Aiming at the problems of the traditional A* algorithm, such as having too many redundant nodes and a long running time, a fusion algorithm combining an improved A* algorithm and segmented second-order Bézier curve was proposed in this paper. Firstly, according to the target node and obstacle distribution, the search neighborhood of the traditional A* algorithm was adjusted from eight to five neighborhoods, eliminating redundant expansion nodes. Secondly, the dynamic weighting factor was introduced on the basis of the evaluation function of the traditional A* algorithm to effectively reduce the expansion nodes and searching time. After that, the key node extraction strategy was used for the paths planned by the A* algorithm to remove the redundant nodes in the optimal paths. Finally, the optimal paths planned by the improved A* algorithm were smoothed using segmented second-order Bézier curves to reduce the path curvature and path length to some extent. Through several experimental simulations, it was found that the length of the optimal paths planned by the fusion algorithm slightly increased compared with the traditional A* algorithm; the optimal paths were smoother, the total number of nodes and running time were significantly reduced, and the optimization effect is more obvious as the size of the raster map increases. Considering the issue of path planning for mobile robots, future research directions will focus on the following four aspects. (1) The strategies to eliminate the "traps" mentioned in Section 6 will be investigated; (2) The fusion algorithm will be applied to the Turtlebot3 robot for realistic path planning simulation experiments, and the characteristics and limitations of the robot will be fully considered; (3) The fusion algorithm will be compared with other recently proposed algorithms to solve the path planning issue of mobile robots, and further verify its effectiveness; and (4) The performance of the fusion algorithm in more complex, U-shaped obstacles as well as dynamic obstacles will be further explored to verify its robustness.

**Author Contributions:** Conceptualization, R.L. and Z.W.; methodology, R.L.; software, Z.W.; validation, R.L., Z.W. and X.L.; formal analysis, X.L.; investigation, R.L.; resources, R.L.; data curation, Z.W.; writing—original draft preparation, Z.W.; writing—review and editing, R.L.; visualization, Z.W.; supervision, N.Z.; project administration, R.L.; funding acquisition, R.L. All authors have read and agreed to the published version of the manuscript.

**Funding:** This research received no external funding.

**Institutional Review Board Statement:** Not applicable.

**Informed Consent Statement:** Not applicable.

**Data Availability Statement:** Not applicable.

**Acknowledgments:** This research was supported by Xiamen Public Technology Service Platform for Digitalization of Industrial Production Enterprises, Xiamen Key Laboratory of Intelligent Manufacturing Equipment, Fujian Province Education and Research Project for Young and Middle-aged Teacher (Granted no. JAT200467, JAT200472), Natural Science Foundation of Fujian Province (No. 2022J011246), and the National Innovation Method Fund of China (No. 2019IM010300). The authors also would like to thank the anonymous referees for their useful comments and constructive recommendations that have improved the quality of the paper.

**Conflicts of Interest:** The authors declare no conflict of interest.

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
