# Peer review of "Fusion Algorithm of the Improved A* Algorithm and Segmented Bézier Curves for the Path Planning of Mobile Robots"

_sustainability, doi:10.3390/su15032483_

Round 1

Reviewer 1 Report

Aiming at the problems of the traditional A* algorithm such as too many redundant nodes and long running time, a fusion algorithm combining the improved A* algorithm and segmented second-order Bézier curve is proposed in this manuscript. The results show that the optimal paths are smoother, and the total number of nodes and running time are significantly reduced. This research topic is very interesting. The manuscript is well-organized. I recommend the acceptance of this work after minor revision.

1, It will be better to clarify the contributions of this work in introduction section. 

2, The process of the fusion algorithm (Section 3) could be described by using pseudocode.

3, It will be better to present the research limitation of proposed algorithm in conclusion section. 

Author Response

Dear reviewer,

Thank you very much for your comments and professional advices. These comments are very helpful and guide us to improve the academic quality of our article. Based on your suggestion and request, we have made corresponding modifications on the revised manuscript, and the details are shown as follows.

Yours Sincerely,

Lai Rongshen

Comment 1: It will be better to clarify the contributions of this work in introduction section. 

The author’s answer: We have added descriptions of main contributions of this article at the end of the introduction section, shown as follows in italics.

“The main contributions of this paper include the following four aspects. (1) A 5-neighborhood search strategy was proposed to optimize the search path, shorten the algorithm execution time, and reduce the redundant nodes. (2) The dynamic weighting factor was added to the heuristic function of the traditional A* algorithm, so that the algorithm can automatically adjust the search focus during the search process, solving the common problem of long algorithm running time of the traditional A* algorithm. (3) A key node extraction strategy was proposed to solve the problem of many redundant nodes in the original global path. (4) A fusion algorithm combining improved A* algorithm and segmented Bézier curves was proposed to smooth the local paths and make the routes smoother and safer.”

In addition, at the end of the introduction section, the overall framework of this paper is added, shown as follows in italics.

“The remainder of this paper is organized as follows. Section 1 describes the improved A* algorithm based on the traditional A* algorithm. Section 2 introduces the basic performance of Bézier curves. Section 3 proposes the fusion algorithm based on the improved A* algorithm and the segmented second-order Bézier curve. Section 4 analyzes the performance of the fusion algorithm. Section 5 discusses the advantages and dis-advantages the proposed fusion algorithm. Section 6 concludes the paper and proposes future research directions.”

Comment 2: The process of the fusion algorithm (Section 3) could be described by using pseudocode.

The author’s answer: According to this suggestion, we have added the pseudocode of the fusion algorithm in Section 3 (Figure 5), shown as follows in italics.

“In this paper, the fusion algorithm combining the improved A* algorithm with segmented Bézier curves is proposed, where the path planning using the improved A* algorithm is implemented first, then the segmented second-order Bézier curves is used to smooth the planned path, so that the planned path has fewer redundant nodes and the path is smoother. The fusion algorithm is described in details as follows, part of the pseudocode is shown in Figure 5, and the specific processes are shown in Figure 6.

Figure 5. The pseudocode of the fusion algorithm.”

Comment 3: It will be better to present the research limitation of proposed algorithm in conclusion section.

The author’s answer: Thank you very much for your suggestion. According to another reviewer’s comment, “If the characteristics and limitations of the robot are considered, it will be more practical. Some discussions can be added”, more practically, there are more than one piece of research limitation (as shown in our current manuscript); therefore, we decide to retain Section 5 “Discussion”, and add other research limitations, shown as follows in italics. In Section 6 “Conclusions”, we mainly focus on the conclusion of this paper and future research directions.

“Moreover, the research mainly considered the randomly generated raster maps, but for some more complex obstacles, such as U-shaped obstacles, it was not analyzed to verify the effectiveness of the proposed method.”

Reviewer 2 Report

1. The contribution of this article should be clearly indicated. There have been many achievements in evolutionary algorithms for path planning. What is the uniqueness between this research and the existing research?

2. Comparison should be strengthened. There is no comparison and discussion with the latest results. DOI: 10.1016/j.knosys.2022.110034.

3. Some more complex U-shaped obstacles can be considered to enhance the effectiveness of the proposed method.

4. If the characteristics and limitations of the robot are considered, it will be more practical. Some discussions can be added.

Author Response

Dear reviewer,

Thank you very much for your comments and professional advices. These comments are very helpful and guide us to improve the academic quality of our article. Based on your suggestion and request, we have made corresponding modifications on the revised manuscript, and the details are shown as follows.

Yours Sincerely,

Lai Rongshen

Comment 1: The contribution of this article should be clearly indicated. There have been many achievements in evolutionary algorithms for path planning. What is the uniqueness between this research and the existing research? 

The author’s answer: We have added descriptions of main contributions and the uniqueness of this research at the end of the introduction section, shown as follows in italics.

“The main contributions and uniqueness of this research include the following four aspects. (1) A 5-neighborhood search strategy was proposed to optimize the search path, shorten the algorithm execution time, and reduce the redundant nodes. (2) The dynamic weighting factor was added to the heuristic function of the traditional A* algorithm, so that the algorithm can automatically adjust the search focus during the search process, solving the common problem of long algorithm running time of the traditional A* algorithm. (3) A key node extraction strategy was proposed to solve the problem of many redundant nodes in the original global path. (4) A fusion algorithm combining improved A* algorithm and segmented Bézier curves was proposed to smooth the local paths and make the routes smoother and safer.”

Comment 2: Comparison should be strengthened. There is no comparison and discussion with the latest results.

The author’s answer: This indeed is one important method to verify the effectiveness and efficiency of the proposed fusion algorithm. It is another one of research limitations added and addressed in Section 5 “Discussion”, and also one of future research directions added in Section 6 “Conclusions”, shown as follows in italics.

“From the comparison perspective, the proposed fusion algorithm was compared with the traditional A* algorithm and the improved A* algorithm, the improved A* algorithm was compared with the traditional A* algorithm, and the comparison results show that, the fusion algorithm is able to optimized the path planning problem and meanwhile illustrate the significance of this research. However, the proposed method should also be further compared with other algorithms proposed most recently to solve the path planning issue of mobile robots.”

“(3) the fusion algorithm will be compared with other algorithms proposed most recently to solve the path planning issue of mobile robots, further verify its effectiveness;”

Comment 3: Some more complex U-shaped obstacles can be considered to enhance the effectiveness of the proposed method.

The author’s answer: According to this comment, we add another one research limitation in Section 5 “Discussion”, and also one of future research directions in Section 6 “Conclusions”, shown as follows in italics.

“Moreover, the research mainly considered the randomly generated raster maps, but for some more complex obstacles, such as U-shaped obstacles, it was not analyzed to ve-rify the effectiveness of the proposed method.”

“(2) the fusion algorithm will be applied to the Turtlebot3 robot for realistic path planning simulation experiments, and the characteristics and limitations of the robot will be fully considered; …… (4) the performance of the fusion algorithm in more complex U-shaped obstacles as well as dynamic obstacles will be further explored to verify its robustness.”

Comment 4: If the characteristics and limitations of the robot are considered, it will be more practical. Some discussions can be added.

The author’s answer: According to Comment 2 and Comment 3, we have add two research limitations in Section 5 “Discussion”, and correspondingly add 2 research directions in Section 6 “Conclusions”.

Reviewer 3 Report

The paper is well-written and well-presented and is accepted for publication. 

Author Response

Dear Sir or Madam,

Thank you very much for your approval of us and the research in the manuscript!

Yours Sincerely,

Lai Rongshen

Reviewer 4 Report

The conventional A* algorithm has problems of a long searching time, an excessive number of redundant nodes, and too many path turning points. The shortest path obtained from planning may not be the optimal route of actual robot movement, and may accelerate robot hardware loss.

 To solve these problems, a fusion path planning algorithm, combining the improved A* algorithm with segmented second-order Bézier curves, is proposed in this paper. the traditional 8-neighborhood search was adjusted to 5-neighborhood search according to the orientation of the target point. The dynamic weighting factor of the heuristic function was introduced into the evaluation function of the traditional A* algorithm. The key node extraction strategy was designed to reduce the redundant nodes of the optimal path. The optimal path planned by the improved A* algorithm was smoothed using segmented second-order Bézier curves.

The paper deals with path planning of mobile robots which is a key topic affecting their functions and performance. The main purpose is to plan an optimal path from the starting point to the target point without collision for a specific obstacle environment. The paper mainly improves the optimal path search time by removing repetitive nodes and turning points.

The strength of this paper is the combination of an improved A* path planning algorithm and a segmented second-order Bessel curves. The traditional 8-neighborhood search is adjusted to 5-neighborhood search which reduces the number of extended nodes and improves the search efficiency. The dynamic weighting factor of the heuristic function is introduced into the traditional evaluation function so that the key nodes are extracted from the planned path to retain key nodes and reduce unnecessary redundant nodes.  The obtained optimal path is then smoothed by segmenting the second-order Bézier curve to get a smoother and collision-free path.  

The procedure followed by the authors is clear and the paper does not require any further improvements in terms of methodology.

The conclusions well summarize the work results and highlight the key findings.

The list of references is adequate and completely summarizes the state of art findings.

Overall, I recommend the acceptance of this paper in its present form

Author Response

(The authors gave the same response as above.)

Round 2

Reviewer 2 Report

Accept